# Stable isotopes show that earthquakes enhance permeability and release water from mountains

Takahiro Hosono [1,2 ✉], Chisato Yamada[3], Michael Manga[4], Chi-Yuen Wang [4] & Masaharu Tanimizu[5]

Hydrogeological properties can change in response to large crustal earthquakes. In particular, permeability can increase leading to coseismic changes in groundwater level and flow. These processes, however, have not been well-characterized at regional scales because of the lack of datasets to describe water provenances before and after earthquakes. Here we use a large data set of water stable isotope ratios ($n = 1150$) to show that newly formed rupture systems crosscut surrounding mountain aquifers, leading to water release that causes groundwater levels to rise (~11 m) in down-gradient aquifers after the 2016 $M_w$ 7.0 Kumamoto earthquake. Neither vertical infiltration of soil water nor the upwelling of deep fluids was the major cause of the observed water level rise. As the Kumamoto setting is representative of volcanic aquifer systems at convergent margins where seismotectonic activity is common, our observations and proposed model should apply more broadly.

[1] Faculty of Advanced Science and Technology, Kumamoto University, 2-39-1 Kurokami, Kumamoto 860-8555, Japan. [2] International Research Organization for Advanced Science and Technology, Kumamoto University, 2-39-1 Kurokami, Kumamoto 860-8555, Japan. [3] Department of Earth Science, Faculty of Science, Kumamoto University, 2-39-1 Kurokami, Kumamoto 860-8555, Japan. [4] Department of Earth and Planetary Science, University of California, Berkeley McCone Hall, Berkeley, CA, USA. [5] School of Science and Technology, Kwansei Gakuin University, 2-1 Gakuen, Sanda 669-1337, Japan. ✉email: hosono@kumamoto-u.ac.jp

Coseismic hydrological changes are widespread and changes in water level after earthquakes are the most frequently documented responses[1,2]. Explanations for changes in water level generally fall into four categories: pore-pressure response to static elastic strain[3,4], fluid migration along seismic ruptures[5–8], permeability changes caused by cracking and seismic vibrations[9–13], and pore-pressure changes in response to liquefaction or consolidation[14–16]. Stable isotope ratios of oxygen and hydrogen in the water molecule ($\delta D$ and $\delta^{18}O$) have been used as a direct water fingerprinting tool to examine the changes between before and after earthquakes[10,17–20]. However, these isotopic studies could not be placed in a regional context because of the lack of good spatial and temporal sampling throughout the watershed. Here we use a comprehensive isotopic dataset obtained from the Kumamoto region (Fig. 1), Japan, to identify and explain subsurface hydrogeological responses to the 2016 $M_w$ 7.0 Kumamoto earthquake.

Groundwater flow in the Kumamoto region generally follows the topographic slope (Fig. 1a). Aquifer systems consist mainly of permeable volcanic pyroclastic deposits, porous lavas, and alluvial deposits of Quaternary ages that overlie hydrogeological basement of relatively impermeable metasedimentary rocks and volcanic rocks of older ages[21]. The two major aquifer systems, an unconfined aquifer (ca. <90 m in depth) and an underlying confined to semi-confined aquifer (ca. 20–200 m thick), are separated by an aquitard (Supplementary Fig. 1). According to previous hydrogeological studies, groundwaters are recharged in the northern and eastern highlands at elevations of ca. 50–200 m (defined as the recharge area, Fig. 1a), then flow laterally south- and westward (lateral flow area), and mostly discharge within 40 years as springs in Lake Ezu at the entrance to the plain area (discharge area)[21–25]. Groundwaters are recharged through soils by precipitation and by river water along the midsection of the Shira River (Fig. 1a). Some nearly stagnant groundwaters remain in the plains and coastal regions to the west of Lake Ezu (stagnant area). Behind these regional groundwater flow systems there are mountain aquifers surrounding the Aso caldera and Kinpo mountains (Fig. 1). These mountain waters discharge as springs both at the base of their respective mountains (defined as mountain foot springs, at elevations ca. 200 m) and at higher elevations (high-elevation mountain springs, ca. 400–620 m). Here, the term groundwater refers to aquifer waters in the regional groundwater flow systems and is distinguished from waters in the mountains which we refer to as mountain aquifer water or mountain spring water.

The destructive inland Kumamoto earthquake sequence began with a large $M_w$ 6.2 foreshock at 21:26 JST on 14 April 2016, followed by the $M_w$ 7.0 main shock at 01:25 JST on 16 April 2016 (Fig. 1b). The earthquake sequence involved strike-slip and normal displacement and revealed many faults and ruptures (Fig. 1b)[26,27]. Groundwater level fluctuations and changes in response to the Kumamoto earthquake were documented in previous reports (see Methods)[6,25]. Briefly, newly recognized normal fault systems, e.g., Suizenji faults[26,27], crosscut groundwater flow systems (Fig. 1b) and led to surface water drawdown into crust deeper than the aquifers, possibly driven by low pressure generated in open cracks (Fig. 1d)[6]. This water-level drop peaked within 35 min after the main shock (Supplementary Figs. 2 and 3; 4.74 m in maximum) because the waters rapidly filled the new cracks. After this initial drop, water levels in these areas tended to recover to the original water levels (Supplementary Figs. 2 and 3). The other notable water level change is a rise in recharge areas (Fig. 1d and Supplementary Figs. 2, 3; a maximum 2.6 m within 45 days and 4.2 m in 1 year of the main shock). In these areas, the initial water level decrease not only recovered but increased to a level higher than the original water table[6]. Hydrological models based on water budgets[28] that considered other possible factors that might cause water level changes (e.g., climate and anthropogenic water extraction and recharge), revealed that the water level increase was triggered coseismically, peaked in 4–5 months, and persisted at least 3 years after the main shock.

Displacements from the Kumamoto earthquake produced extensional strain over the study area except in the eastern mountains[6]. Moreover, the groundwater levels initially dropped after the main shock in both recharge and discharge areas (Fig. 1d and Supplementary Fig. 2). It is thus difficult to explain the observed water level rise by pore-pressure increase in response to crustal strain changes[6]. Continuous water level increase after earthquakes has been reported elsewhere, and has been explained by increasing contributions of new waters owing to permeability enhancement[10,13,29]. Further, it has been reported that the discharge rates of the Shira River increased near the mountains after the main shock[6]. It is therefore possible that increased permeability in the upstream area is responsible for the observed groundwater level rise in downgradient groundwater flow systems. Previous studies documenting coseismic water level rise in response to permeability enhancement[12,29] proposed three possible new water pathways including soil porewater infiltration from the unsaturated zone[11], groundwater mixing among different aquifers through new cracks[30–33], and increased contributions from aquifers sourced in the surrounding mountains[10,13,33]. However, comprehensive isotopic assessment to identify these sources, including the contribution of deep fluids and liquefaction, has not yet been achieved.

This present study identifies the origin of water level changes based on isotopic fingerprinting using a large number of samples ($n = 1150$) of all possible water sources from the regional watershed (Fig. 1c, see Methods for sampling strategy), comparing data before and after the earthquake (e.g., Figs. 2, 3 and Supplementary Tables 1, 2). Our results are then used to discuss the processes that led to groundwater level rise in response to the 2016 Kumamoto earthquake.

## Results

**Isotopic compositions of waters before the earthquake.** Figure 2 shows the isotopic compositions of waters before the earthquake, including precipitation, soil porewaters, river waters, springs, and groundwaters (see Fig. 1c for their sampling locations). In Fig. 2, soil porewaters obtained from recharge areas plot slightly to the right of the local meteoric water line for the high-water season (April–September, see Methods). We thus suggest that these waters are recharged by precipitation during the high-water season and were partly evaporated before infiltration (see evaporation trend shown by the dotted arrow in Fig. 2)[23]. In contrast, the samples of mountain springs and the Shira River plot to the left of the local meteoric water line. In addition, the high-elevation mountain springs and the Shira River waters generally show more depleted isotopic signatures than mountain foot springs (Fig. 2), reflecting their higher recharge elevations (see the altitude effect trend shown by solid arrow in Fig. 2).

The isotopic compositions of groundwater samples collected before the earthquake plot along the local meteoric line for the high-water season and within a compositional field surrounded by that of soil porewaters, mountain spring waters, and the surface river waters (Fig. 2). These isotopic data imply that groundwaters are recharged by precipitation between April and September and could be mixture waters of soil porewaters, mountain aquifer waters, and the Shira River waters. The isotopic signatures of waters are the same in both aquifers (Supplementary Fig. 4a) although there are fewer samples from unconfined aquifers due to the smaller number of monitoring wells (Fig. 1c).

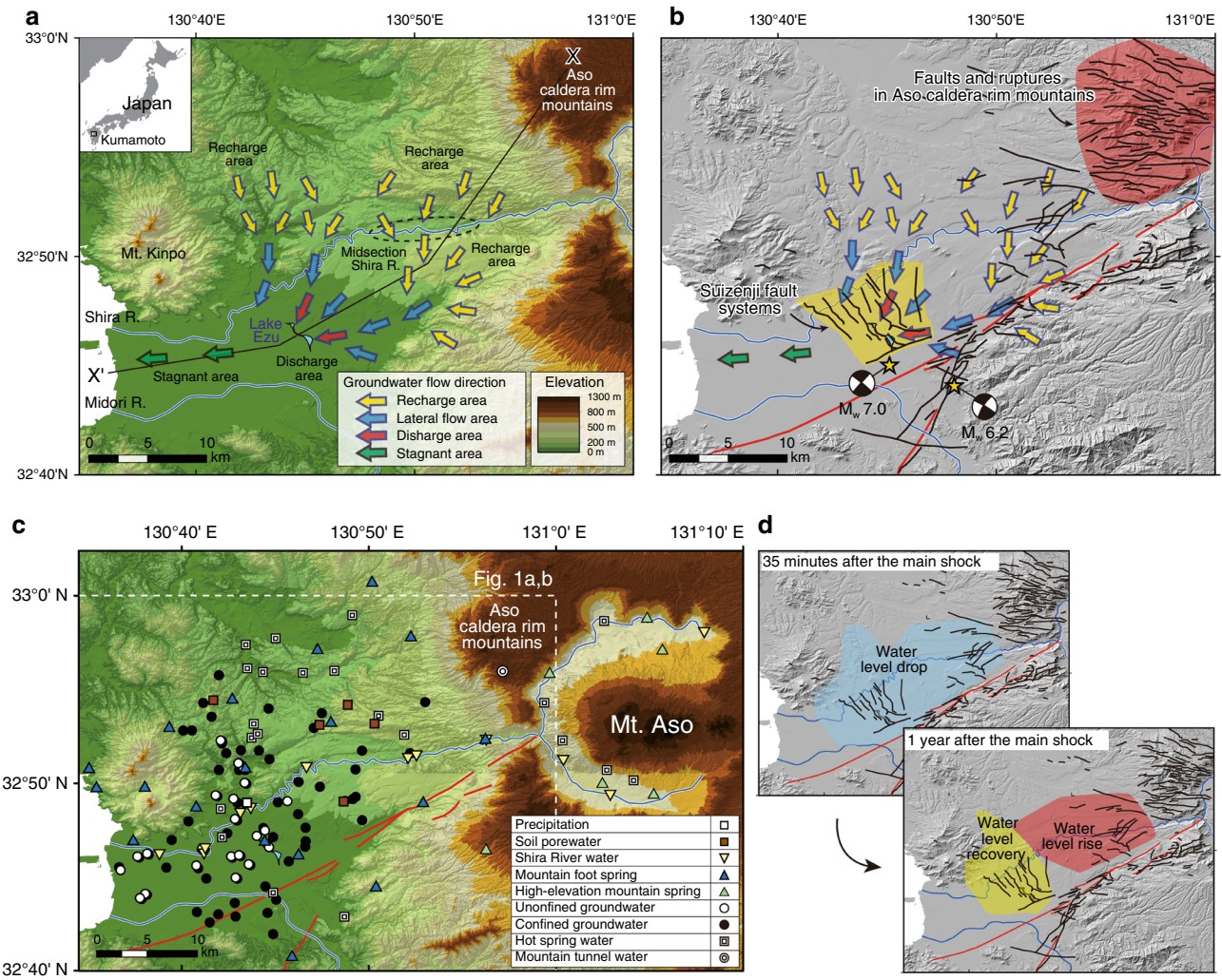

**Fig. 1 Hydrogeology and seismotectonics of Kumamoto. a** Map showing original hydrological systems in the study area before the 2016 Kumamoto earthquake[6]. Flow directions in each area of the confined aquifer are shown with arrows of different colors. **b** Seismotectonic structures crosscutting regional groundwater flow systems after the 2016 Kumamoto earthquake. Distributions of the foreshock and main shock epicenters (stars) with their mechanisms (beach ball plots), pre-existing fault systems (red lines), and locations of newly recognized ruptures (black lines) are from refs. [6, 21]. **c** Sampling locations for various waters used for isotopic comparisons. **d** Cartoon showing the areas where groundwater level changed 35 min and 1 year after the main shock (see Supplementary Fig. 2 for detailed water level change distributions). Hydrogeological cross-section X–X′ is shown in Fig. 4.

A contribution of mountain aquifer waters to down-gradient regional groundwater flow systems has not been documented in previous studies; our results suggest that groundwaters are recharged partially by mountain aquifer waters.

Isotopic compositions of groundwaters are stable regardless of the season, with annual variabilities of <±0.12‰ and ±0.5‰ for $\delta^{18}O$ and $\delta D$, respectively, based on monthly samplings in groundwater discharge areas[34] ($n = 70$, Supplementary Table 3). As expected, isotopic compositions of spring water samples did not show systematic changes due to different sampling months (Supplementary Fig. 4b). In addition, groundwater samples collected during two different months show overlapping isotopic compositions for both aquifers (Supplementary Fig. 4c, d). However, there are some samples from the beginning of the low-water season (October to December) with greater $\delta^{18}O$ than those later in the low-water season (January to March) (Supplementary Fig. 4c, d). This may be due to a lack of sufficient samples or may imply a relatively higher contribution of soil porewaters transported through preferential flow pathways in the rainy season (June and July, see Methods), as may be typical in recharge areas[35]. Isotopic characterization of water prior to the

earthquake enables us to assess coseismic isotopic changes and to identify the origins of waters that caused groundwater level to rise after the main shock.

**Isotopic compositions of waters after the earthquake.** In Fig. 3 and Supplementary Fig. 5, the most remarkable post-seismic isotopic changes are observed for groundwaters: the compositional range of samples changed from wider (shown by the field surrounded by black line in Fig. 3b–d and Supplementary Fig. 5) to a more narrow range that is more similar to mountain foot spring waters before the earthquake (most red-circle symbols plot within a field surrounded by the blue line), regardless the sampling season, aquifer type (confined vs. unconfined), and area of aquifers (except the stagnant area; Fig. 1a). The samples of stagnant groundwaters fall into a narrow compositional field along the high-water season local meteoric water line for both pre- and post-seismic periods. Both mountain foot and high-elevation spring waters changed in $\delta D$ and $\delta^{18}O$ toward slightly depleted compositions, while those of river waters did not show significant changes after the earthquake (Fig. 3a). The cause of

post-seismic isotopic changes for mountain springs has been discussed elsewhere[36].

Isotopic compositions of the hot spring waters obtained from deep water reservoirs (180 to 1300 m below the ground surface) show a wide range (Fig. 3a) and are divided into four groups: compositions identical to high-elevation mountain springs, soil porewaters, water with the highest δD (−36.5‰) and δ18O (−1.84‰), and waters with relatively lower δD but higher δ18O (−1.84‰). The hot springs for the first two groups have meteoric water origins similar to high-elevation mountain aquifers and waters that infiltrate recharge areas, respectively. The isotopic signatures of waters for the third and fourth groups, which are located near the coast and under the northeastern recharge areas respectively, are mixtures of sea water with high δD and δ18O (≈0‰) and deep geofluids that experienced high temperature water-rock interaction resulting in δ18O isotopic shifts towards enriched compositions leaving δD unchanged[37], respectively. Here we use the compositional field of hot spring waters of the fourth group as a proxy for typical deep fluids that may contribute to groundwater level rise after the earthquake. The compositional ranges of data before the earthquake were used for mountain foot and high-elevation mountain springs as isotopic references for fingerprinting assessment (Fig. 3 and Supplementary Fig. 5).

Figure 3b–d and Supplementary Fig. 5 include groundwater samples collected from different seasons. To eliminate any possible seasonal effects, only samples collected in October and November when the groundwater level is highest (ca. 3 months after the rainy season, see Methods), are shown in Supplementary Fig. 6. In addition, observed isotopic changes are cross-checked by using the results from the other sampling campaigns during September 2015 and March 2017 when measurements were made with a different analytical device and measured by other institutions (Supplementary Fig. 7 and Supplementary Table 2). Results of these comparisons (Supplementary Figs. 6 and 7) confirm that the observed isotopic changes were not caused by seasonal differences, difference in sampling years, nor analytical methods. Consequently, our measurements require that there were contributions of new waters derived from different pathways than the original hydrogeological systems. These may have caused post-seismic water levels to rise (recharge by lateral flow) and recover (lateral flow to discharge areas); see Supplementary Fig. 2.

**Origin of additional water.** The most remarkable result of our isotopic comparisons is that the composition of groundwaters changed from resembling a mixture of multiple sources into a composition with a signature more similar to mountain foot spring waters. Water compositions did not change towards those of soil porewaters, river waters, and deep fluids (Fig. 3b–d and Supplementary Fig. 5). This result implies an increased contribution of water from mountain aquifers.

If the seismic ruptures crosscut aquitards between the unconfined and confined aquifers and waters from these two aquifers mixed with each other, isotope ratios should reflect that mixing. However, analyzed isotopic ratios for both aquifers generally changed toward compositional fields that are different from average groundwater compositions (Fig. 3b–d and Supplementary Fig. 5). Continuous water level rise in recharge areas and water recovery around the lateral flow and discharge areas are thus most simply explained by an increasing contribution of mountain aquifer waters[10,13] from above the water recharge areas. This inference supports the hypothesis derived from hydrological analyses both for surface and subsurface waters[6,28] and hydrochemical signatures that suggest mixing of diluted mountain aquifer waters[38].

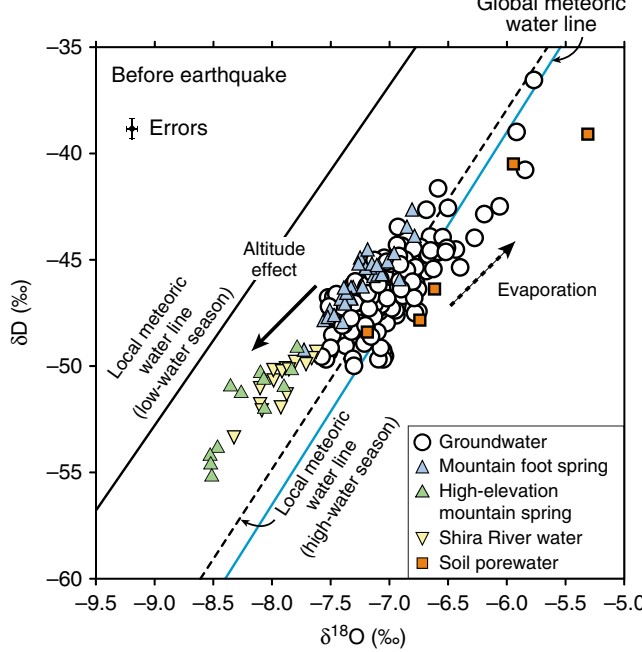

**Fig. 2 Isotopic compositions of waters before the earthquake.** Oxygen and hydrogen stable isotope ratios characterizing the original hydrological systems in Kumamoto. Samples from different seasons are included. Two local meteoric water lines including high-water (during April to September, $n = 70$) and low-water (during October to March, $n = 65$) seasons are shown in the figures using monthly precipitation data (see Methods). The global meteoric water line and isotopic evolution trends due to altitude effects and evaporation are from refs. [55-57]. Error bars in the figure represent analytical precision.

To more precisely identify the origins of new waters from mountain aquifers, recharge elevations of mountain spring waters are characterized isotopically. Here, analyzed mountain spring waters are classified into two types, mountain foot springs (3 to 191 m, above sea level) with relatively enriched δD (−49.2 to −42.6‰) and δ18O (−7.79 to −6.79‰) and high-elevation mountain springs (407–620 m) with relatively depleted δD (−55.3 to −49.1‰) and δ18O (−8.76 to −7.79‰) (Figs. 2, 3). Based on previously reported regressions for determining water recharge elevation[39]

$$\delta D = -0.0164\,h - 39.153,$$

where δD and h (m, above sea level) represent δD values of spring waters and their recharge elevations, respectively, the recharge elevations for each type of spring are estimated as ca. 210–613 m and ca. 607–985 m, respectively (Fig. 4a). Figure 3b–d shows that isotopic compositions of almost all groundwaters after the earthquake approach those of mountain foot springs with intermediate recharge elevations (210–613 m) on the western Aso caldera rim and Mt. Kinpo.

Recent analyses using chemical[38] and microbiological tracers[40] have detected in some localized areas an increased contribution of soil porewaters, river waters, and deep fluids after the 2016 Kumamoto earthquake. For instance, analysis of a long-term chemical monitoring dataset revealed post-seismic $NO_3^-$ increase in groundwaters in recharge areas that is attributed to enhanced percolation of soil porewaters into aquifers from agricultural fields triggered by seismic vibrations[38]. The same study also revealed a post-seismic increase in the contribution of deep fluids to surface aquifers based on

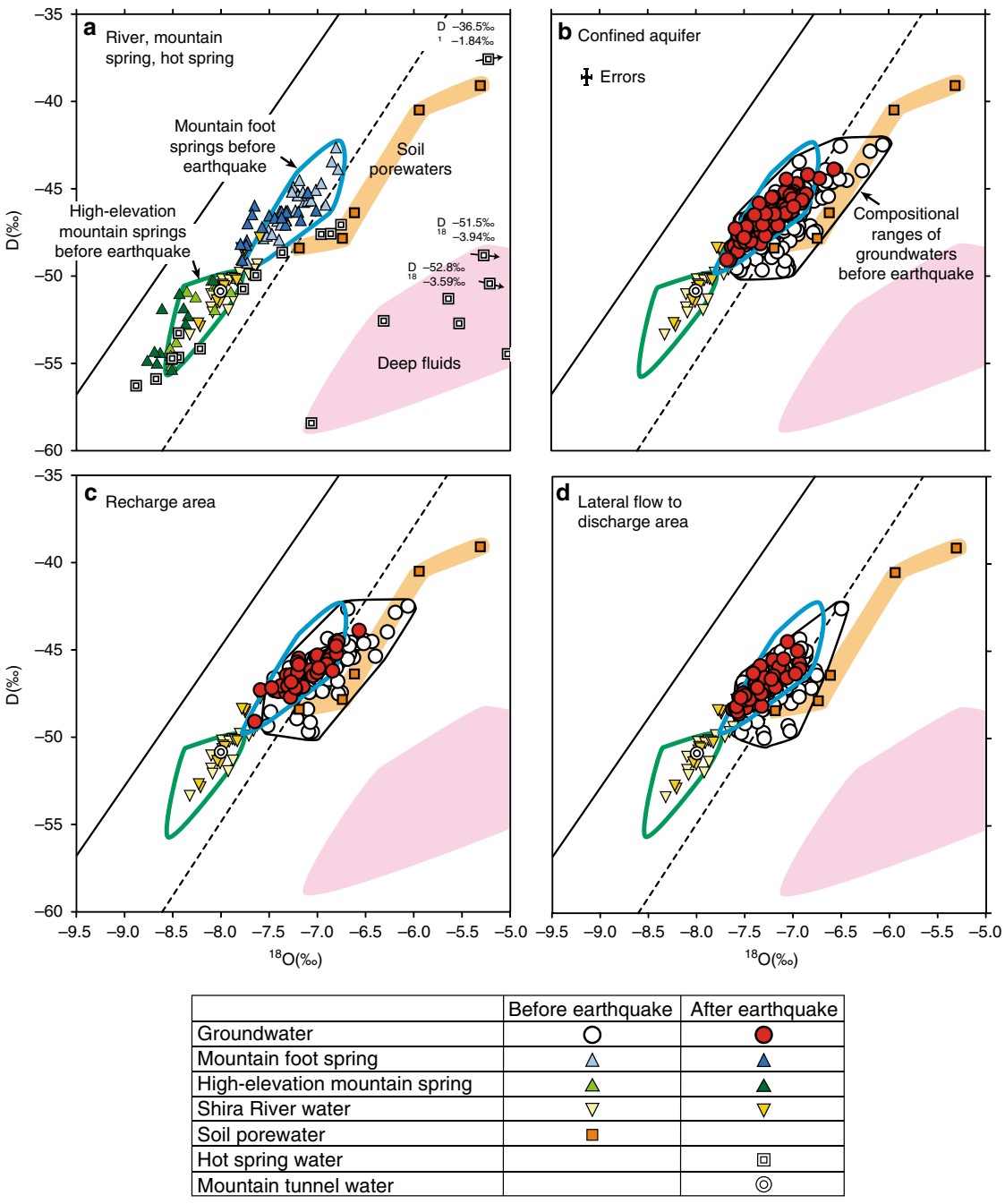

**Fig. 3 Coseismic changes in stable isotope ratios. a** Oxygen and hydrogen stable isotope ratios showing compositional changes before (April 2011 to July 2011) and after (August 2016 to May 2017) the main shock for river and spring waters for the samples from various seasons. Compositions of hot spring waters and mountain aquifer water from ongoing tunnel construction for the samples collected after the main shock are also plotted. Springs (blue and green triangles) and river (yellow triangle) water samples obtained after the earthquake are shown in darker colors than samples from before the earthquake. **b** Compositional changes before (November 2009 to November 2011) and after (June 2016 to December 2017) the main shock for confined groundwaters for the samples from various seasons. **c**, **d** Compositional changes of groundwaters for recharge and lateral flow to discharge areas, respectively. Samples from all seasons for both aquifers (unconfined and confined aquifers) are plotted. The two local meteoric water lines are the same as in Fig. 2. In b–d, groundwater samples after the earthquake are shown in red, while those before the earthquake are shown in white. Errors are shown in Fig. 3b.

geochemical tracers, in particular $Cl^-$, $SO_4^{2-}$, and B, as has been documented in many other instances[41–46]. This phenomenon occurs near the epicentre of earthquakes and in geothermal regions[38]. In addition, the microbiology in aquifer waters dramatically changed after the earthquake with increased exogenous microorganisms found in deep groundwaters[40] such as *Propionibacterium acnes*, which originally inhabited the surface environment. Mixing of river waters into deeper aquifer

systems is thus likely. These hydrochemical and microbial anomalies continued for at least 2 years after the earthquake. However, these effects are not reflected in a change of groundwater isotopic compositions. Thus, their contributions are negligible in terms of water volume and unlikely to cause the observed groundwater level rise and recovery in regional groundwater flow systems, except for the river water contribution that will be discussed later.

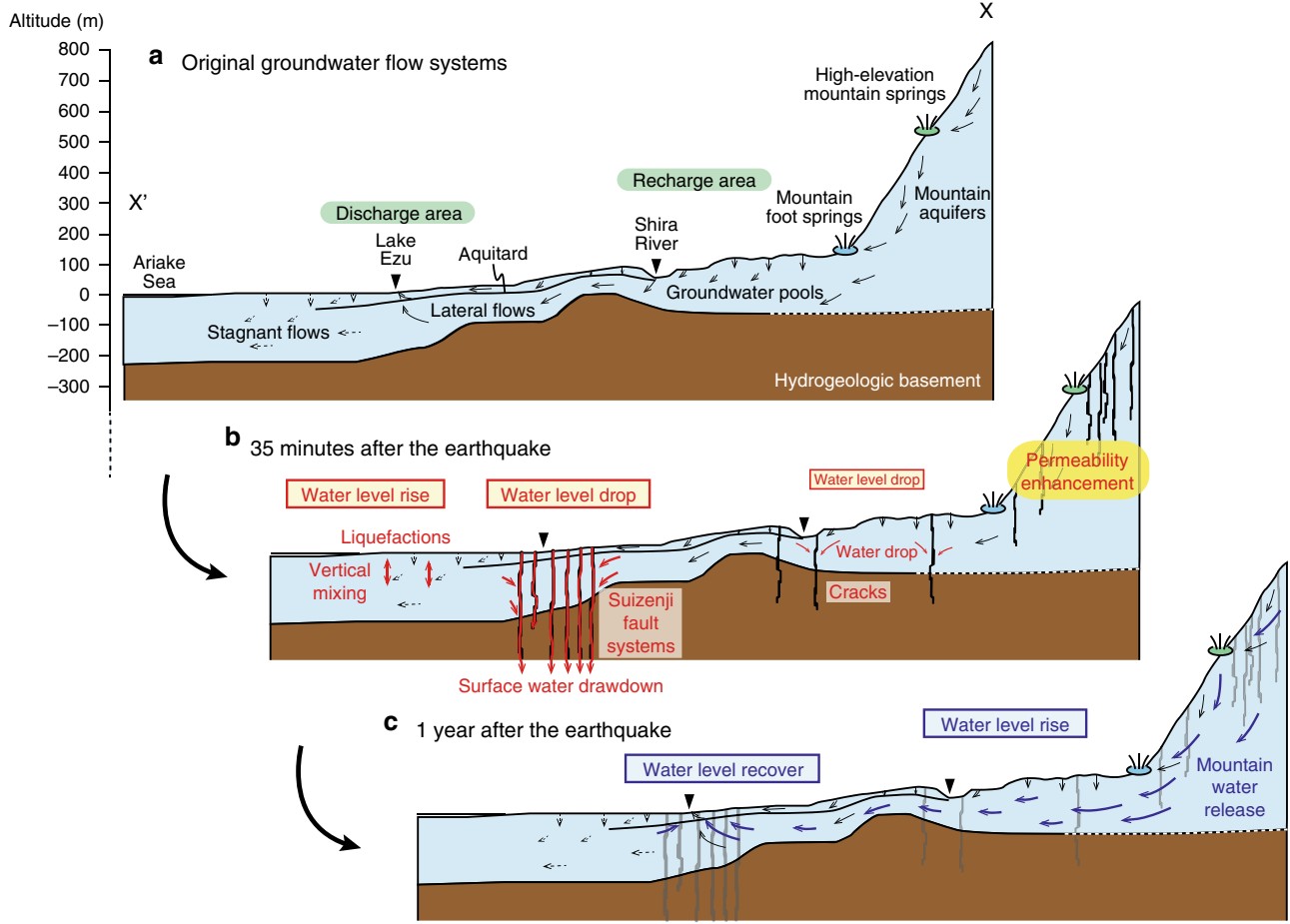

**Fig. 4 Cartoons showing coseismic hydrogeological changes. a** Schematic cross-section X–X' (see Fig. 1a for its location) showing original hydrogeological systems in the study area. **b, c** Cross-sections showing coseismic hydrogeological changes 35 min and 1 year after the main shock of the 2016 Kumamoto earthquake, respectively. Coseismic groundwater drawdown mechanisms are after ref. [6].

## Discussion

Detailed descriptions of newly recognized surface ruptures (Fig. 1b)[26] showed that the formation of the Suizenji fault systems (Fig. 1b) played an important role in inducing groundwater drawdown immediately after the earthquake (Fig. 4b) prior to the subsequent water level rise[6]. Similar concentrated rupture systems are present in the eastern caldera rim mountains (Fig. 1b), and these may provide the pathways to enhance flow from mountain aquifers to downslope groundwater systems after the earthquake[13]. This hypothesis is consistent with the observation that some springs in the mountains became dry after the earthquake[36]. However, the high-elevation mountain springs and the sample of mountain aquifer water directly obtained from ongoing tunnel construction under caldera rim mountains (see Fig. 1c for its location), are not isotopically identical to the hypothesized additional waters (Fig. 3b–d). Rather, our isotopic analysis suggests that fracture systems near the base of the western Aso caldera rim and Mt. Kinpo, were the dominant pathways for new waters from mountain aquifers (Fig. 4c).

The water redistribution from mountains to downslope aquifers identified by isotopic fingerprinting is most obvious in the eastern recharge area (Fig. 3c and Supplementary Fig. 8) where the most significant abnormal water level rise was observed (Supplementary Figs. 2 and 3): isotopic compositions of groundwaters changed after the earthquake towards the compositional field of mountain foot spring waters in the vicinity of this area, suggesting that mountain water was released and mixed with aquifer waters in the recharge areas. Moreover, mountain

foot spring waters changed their isotopic compositions to more depleted values (Supplementary Fig. 8), implying a contribution of mountain waters with higher elevations (Fig. 4c). This is further evidence of permeability enhancement of mountain aquifers.

The response of mountain waters appeared within 1 day as increased river and spring discharges[6,47] and abnormal groundwater level rise in recharge area (Supplementary Fig. 3). The water levels in this area continuously rose (Supplementary Figs. 2, 3) and peaked 4–5 months after the main shock with a maximum abnormal water level rise of ~11 m[28]. In general, the recharge areas consist of major groundwater reservoirs in Kumamoto, locally called groundwater pools (Fig. 4a), and have the highest seasonal water level fluctuations of up to ~10 m[25]. Thus, a large volume of coseismically-released mountain waters can be transferred to those groundwater pools in the recharge area.

The isotopic results (Fig. 3d) further suggest that water in the groundwater pools was then transported down slope and led to water level recovery by lateral flow to recharge areas over the course of the annual hydrologic cycle (Supplementary Fig. 2). This isotopic constraint requires a faster flow than that suggested by water residence times of a few to a few tens of years estimated by chemical age tracers[25]. Aquifers in recharge and discharge areas, and partly in lateral flow areas, may also have experienced permeability enhancement from the formation of rupture systems (Fig. 1b)[26]. Previous studies identified faster flow only during heavy rain[35]. We can therefore assume that the lowered water levels due to coseismic groundwater drawdown recovered by the contribution of large amounts of mountain waters (ca. $10^8$ m$^3$)[47]

that were transported through these preferential subsurface pathways. A few groundwater samples collected within this fault zone changed their isotopic compositions toward those of Shira River water samples (Supplementary Fig. 9). Although such signals are the exception to the general trend, they imply that the surface waters could travel along preferential pathways, explaining the decline of Shira River water levels near the fault zone during the first 12 h of the main shock[6] and may have partially contributed to subsequent water level recovery.

Significant isotopic alterations were not observed for the waters in stagnant areas (Supplementary Fig. 5b). Some wells in unconfined aquifers in these areas showed coseismic water level rise immediately after the main shock (Supplementary Fig. 2), which has been attributed to pore-pressure increase in response to liquefaction (Fig. 4b)[6]. Vertical water mixing within these aquifer systems will not cause isotopic compositional changes.

Our large isotopic datasets, covering the time before and after the earthquake, allow us to elucidate the origins and processes of post-seismic groundwater level rise and recovery that are caused by water release from mountain aquifers triggered by permeability enhancement after the 2016 Kumamoto earthquake. With these findings and previous work describing a short-lived initial water level drop[6], we identify two major stages of coseismic regional changes (Fig. 4). First, surface waters and groundwaters dropped (4.74 m maximum) immediately (within 35 min) after the main shock along newly formed cracks in Suizenji fault systems (Fig. 4b). Second, water levels rose in recharge areas in response to mountain waters being released from mid-elevation areas by permeability enhancement. This water flowed to discharge areas through preferential pathways leading to subsequent water level recovery around the Suizenji fault area (Fig. 4c). Numerical simulations involving permeability changes have reproduced these regional flow changes using a physically-based integrated watershed modeling tool[47]. In recharge areas, water level rise anomalies still remain 3 years after the main shock, hypothesized to be because of persistent permeability increase in mountain aquifers. However, further downslope, water levels almost recovered over the annual hydrological cycle by lateral flow to discharge areas. The results of this study provide a hydrogeological framework to understand other environmental changes including water temparature[48], chemistry[38], microbiology[40], and water supply security[49].

This study illustrates how large crustal earthquakes may alter regional hydrological systems hosted in rocks of volcanic origin. Their often high permeability and storage lead to widespread use for groundwater supplies. Global geographic overlap between volcanic and seismotectonic activity suggests that similar coseismic hydrological changes can be anticipated in volcanic arcs worldwide.

## Methods

**Hydrogeological setting**. Kumamoto has a humid monsoon-dominated climate and shows four distinct seasons (http://www.data.jma.go.jp/gmd/cpd/longfcst/en/tourist.html). The annual average precipitation in the Kumamoto and Aso areas are 1.99 and 2.83 m y$^{-1}$ with average temperatures of 16.9 and 12.9 °C, respectively (data sources: 1980–2010, http://www.jma.go.jp/jma/menu/report.html for Kumamoto; 1981–2010, https://weather.time-j.net/Climate/Chart/asootohime for Aso). In Kumamoto, ~75% of annual precipitation occurs from April through September (we call this season the high-water season), while the remaining 25% occurs from October through March (low-water season). The rainy season (June and July) accounts for about 40% of the total annual precipitation. The Kumamoto groundwater area is bounded by the Shira River watershed to the north, the Midori River to the south, the outer rim of Aso caldera (highest peak: 1154 m) to the east, and the Ariake Sea and Kinpo Mountain (665 m) to the west (Fig. 1a)[21]. The Aso caldera watershed (380 km$^2$) is hosted within a large caldera (25 × 18 km, Fig. 1c). The ring-shape caldera rim forms the watershed divide and the central volcanic mountains (highest peak: 1592 m) are situated in the central part of the caldera (Fig. 1c).

The groundwater flow systems are briefly described in the main text. More detailed topography, geology, hydrology, seismotectonics, and sociology are provided in refs. [6,21,25,50]. In addition, detailed groundwater fluctuations with and without seismic effects, evaluations of post-seismic water level changes considering these water level fluctuations, and changes in ground level induced by the earthquake are documented in previous studies[6,25,28]. In general, groundwater levels show seasonal changes and are the lowest during April and May, and highest during October around 3 months after the rainy season. The water level changes within 45 days after the main shock (Supplementary Figs. 2, 3) can be regarded as co-seismic changes because seasonal fluctuations are much smaller than observed water level changes during April and May 2016 (Hosono et al.[6]). It has been reported that liquefaction occurred predominantly in the plains and coastal areas[51] where near-stagnant groundwaters are found (Fig. 1a) in soft marine clay sediments (generally ~60 m thickness). In these areas, water level rose immediately after the main shock for some wells in unconfined aquifers (Supplementary Fig. 2).

**Samples**. A total of 872 samples were collected for isotopic analysis and characterized for hydrological systems before the earthquake (see Fig. 1c for their sampling locations): 135 monthly precipitation samples during 2005 and 2016 (Okumura et al.[23]), 500 soil porewater samples from five borehole cores from the unsaturated zone in the recharge area during 2012–2014 (Okumura et al.[23]), 15 surface water samples from the Shira River during April 2011 and July 2011, 45 mountain spring water samples during April 2011 and July 2011 (Ide et al.[36]), 43 unconfined groundwater samples from municipal and national monitoring wells (the same wells or at the same locations where water levels are monitored) during November 2009 and November 2011, and 134 confined groundwater samples from municipal and national monitoring wells during November 2009 and November 2011.

All stable isotope datasets with their sampling locations and dates are provided in Supplementary Tables 1 and 2 except for precipitation and soil porewater data which have been provided in a previous study[23]. Precipitation samples were collected on the roof of Kumamoto University's building and data were used to define the local meteoric line (e.g., Fig. 2). In general, vertical soil porewater profiles (measured at 10 cm intervals in cores) display isotopic fractionations reflecting seasonal variability in precipitation[23]. Therefore, average values for the top 10 m (100 samples for each core) were plotted for five cores in Fig. 2 as a proxy of recharge waters in the recharge areas (Fig. 1c). All river, spring, and groundwater samples are the same samples used for other isotopic and geochemical measurements in previously published articles[21,52–54].

In total, 201 river, spring, and groundwater samples were collected after the main shock during June 2016 and December 2017 at the same sampling sites where samples had been collected before the earthquake: the river waters ($n = 11$) were sampled in August 2016 to April 2017, the spring water samples ($n = 30$) were collected in October 2016 and March–May 2017 (Ide et al.[36]), whereas the groundwaters ($n = 160$) were sampled during four campaigns in June-August 2016, October–November 2016, March–May 2017, and November–December 2017. In addition, 23 hot spring water samples were collected over the study area from 180 to 1300 m deep boreholes in July–August 2018 to test for the possibility of deep fluids contributing to the water level changes. One mountain spring water sample from ongoing tunnel construction (182 m below ground surface, 582 m above sea level) was also collected in October 2017 (Fig. 1c)[36].

**Analytical procedures**. All samples except soil porewaters were collected on-site and stored in a 20-ml glass vials, whereas soil porewaters were sampled in the laboratory after extraction from core soils by centrifugation under pF = 4.2 (Okumura et al.[23]). All sample vials were tightly capped and stored in the dark and at room temperature. Hydrogen and oxygen stable isotope ratios were determined by a continuous-flow gas-ratio mass spectrometer at the Kumamoto University (Delta V Advantage, Thermo Fisher Scientific, USA). Based on replicate measurements of standards and samples, the analytical precisions (standard deviations) for δD and δ$^{18}$O were better than ±0.5‰ and ±0.05‰, respectively. Both isotope ratios are expressed in delta notation (δ) in per mill unit (‰) with respect to international standards of Vienna Standard Mean Ocean Water.

Observed coseismic isotopic changes are cross-checked by the results from the other sampling campaigns with data analyzed with another analytical machine. Groundwater samples were collected three times from the study area during September 2015, August 2016, and March 2017. Collected samples were analyzed for δD and δ$^{18}$O using cavity ring-down spectroscopy (Piccaro, L2120i and L2130i) installed at the Research Institute for Humanity and Nature, Japan. The analytical precisions of δD and δ$^{18}$O for this instrument are better than ±0.5‰ and ±0.1‰, respectively. Sampling locations and analyzed data are shown in Supplementary Fig. 10 and Supplementary Table 2, respectively.

## Data availability

All isotopic data presented in Figs. 2 and 3 and Supplementary Figs. 4, 5, 6, 8b, and 9b are included in this article as Supplementary Table 1, except those for precipitation and soil porewater samples whose data are provided in Okumura et al.[23]. Source data used for illustrating Supplementary Figs. 7 and 10 are listed in Supplementary Table 2. Groundwater level data shown in Supplementary Figs. 2 and 3 are listed in Hosono et al.[6]. Additional information is available from the corresponding author upon resonable request.

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

## Acknowledgements

The government office of Kumamoto City, Kumamoto City Waterworks and Sewerage Bureau, Kumamoto Prefecture, and Ministry of Land, Infrastructure, Transport and Tourism kindly supplied opportunities for water samplings. Dr. H. Nakata, Dr. J. Shimada, Dr. K. Ichiyanagi, Dr. K. Ide, Mr. T. Tokunaga, Mr. K. Fukamizu, Mr. N. Sugimoto, Ms. E. Ishii, and Ms. M. Hashimoto helped us for sampling, discussion and analyzing the water stable isotope ratios. T.H. was supported by the 2016 JST J-Rapid Program (leader, Dr. H. Nakata, Kumamoto University), JSPS Grant-in-Aid for Scientific Research B (17H01861), JSPS Fostering Joint International Research A (19KK0291) and SUNTORY Kumamoto groundwater research project. T.H. wishes to thank working group members of Japanese Association for Groundwater Hydrology for fruitful discussion. M.T. was supported by the JSPS Grant-in-Aid for Scientific Research C (17K00527), Kwansei Gakuin University Joint Research in 2016, and Joint Research Grant for the Environmental Isotope Study of Research Institute for Humanity and Nature for isotopic analysis. M.M. and C.Y.W. were supported by the US National Science Foundation 1615203.

## Author contributions

T.H. conceived the idea and wrote the manuscript; C.Y. and M.T. analyzed hydrogen and oxygen isotope ratios; M.M. and C.Y.W. deepened discussions and writings.

## Competing interests

The authors declare no competing interests.
