## [Peer Review File · Nature Communications]

Reviewers' comments:

Reviewer #1 (Remarks to the Author):

Review by Alasdair Skelton of Earthquakes release water from mountains submitted to Nature Communications by Takahiro Hosono and co-authors.

What are the major claims of the paper?

The submitted manuscript builds from the published work of Hosono et al. (2019) in which changes of groundwater levels are shown to have occurred in response to the 2016 Mw 7.0 Kumamoto earthquake. In the submitted manuscript, the authors use an extensive stable isotope dataset for oxygen and hydrogen ($n = 1150$) to show that these changes were caused by a release of water from mountain aquifers in response to co-seismic rupturing.

Hosono, T., Yamada, C., Shibata, T., Tawara, Y., Wang, C. -Y., Manga, M., Rahman, A. T. M. S. & Shimada, J. Coseismic groundwater drawdown along crustal ruptures during the 2016 Mw 7.0 Kumamoto earthquake. *Water Resources Research* 55(7), 5891–5903 (2019).
doi.org/10.1029/2019WR024871

Are they novel and will they be of interest to others in the community and the wider field?

This claim is novel for the reason given by the authors “because of the lack of datasets to describe water provenances before and after earthquakes”. The claim is of interest not only for researchers studying earthquake hydrology but also with respect to water security, a key issue for earthquake hazard mitigation.

Is the work convincing, and if not, what further evidence would be required to strengthen the conclusions?

The work is convincing and the conclusions are largely supported by the data. There are, however, some areas of ambiguity where improvements are necessary, as well as some issues related to the disposition of the manuscript, both of which will be discussed below.

On a more subjective note, do you feel that the paper will influence thinking in the field?

The manuscript will influence thinking in the field in that it elegantly demonstrates how stable isotopes can be used as a powerful and widely applicable tool for understanding hydrological and hydrochemical changes coupled with earthquakes.

Major comments

I have two major comments, one concerning interpretation of the data and one concerning the disposition of the manuscript:

Concerning interpretation of the data: Data in Fig. 2 are used to argue that “the isotope ratios of groundwaters changed ... from their original compositions towards compositional fields of mountain spring waters ...” (lines 125-128). What is actually observed on the figure is a clustering of isotopic data towards the compositional field of mountain foot spring waters. Prior to the earthquake a good proportion of groundwater data overlapped mountain spring data. This difference does not

undermine the conclusions of the study: the wider spread of isotopic values prior to the earthquake probably affects mixing of mountain spring water, soil pore waters and surface river waters. After the earthquake, mountain foot spring water clearly dominates the system, supporting the authors' conclusions.

Concerning the disposition of the manuscript: Lines 55-80 are not results of this study: they are results of the study published in *Water Resources Research* by Hosono et al. (2019). These "results" should therefore form part of the previous section which should be entitled "Introduction" or similar. The results for the manuscript are described in lines 82-123, so this section should be entitled "Results". These new results are discussed in lines 125-194, so this section should be entitled "Discussion". The conclusion and wider significance of the study are presented in lines 197-216, so this section should also be entitled accordingly.

Minor comments

Lines 85-86: I suggest rewording "plot along near the local meteoric line during high water season (April to September including rainy season) but to the right to this line" to "plot slightly to the right of the local meteoric water line for the high-water season (April to September including rainy season)."

Lines 88 and 91: Here you refer to isotopic shifts due to partial evaporation and recharge elevation. This would be easier for the reader to follow by adding corresponding arrows to show the directions of these shifts in Fig. 2.

Line 89: I suggest replacing "to the local meteoric water" with "of this".

Line 94: I suggest replacing "during" with "for the".

Line 97: I suggest replacing "are" with "could be". It is clear that that "groundwaters in the study area are recharged by precipitation during April to September" but unclear that mixing of all three of soil porewaters, mountain aquifer waters and the Shira River waters must have occurred.

Lines 98-99: You state that the isotopic signatures of water are the same in unconfined and confined aquifers. I am not sure how the reader is expected to see this from the data as it is presented.

Line 104: Please replace "mountain spring waters" with "mountain foot spring waters"

Line 139: Please replace "Fig. 4a" with "Supplementary Fig. 4a".

Reviewer #2 (Remarks to the Author):

Key results:

The paper attempts to use a new approach (isotope analysis) to determine the source of water that has resulted in a groundwater level rise post-earthquake. The large number of samples available means that this is quite a novel approach.

Validity:

In general I consider that, in an attempt to keep the paper brief, there is a lot of background information left out, which makes interpretation of the paper difficult. The structure also makes it difficult to follow. As it is, I would say that the information presented is not sufficient to justify the proposed conclusions, but it would be interesting if the data could support the conclusions. There

are also a number of specific issues:

The hypothesis is that an increase in permeability has resulted in a groundwater level rise. Increases in permeability cause a decrease in groundwater level (Darcy's Law). If the hypothesis is that there is enough additional input to overcome increased output, or is it that permeability has increased in some areas and not in others, this needs to be clearly explained.

The figures are unclear about what they are showing. References to the figures say that they show something, but it is difficult to understand the point that is being made. For example, Fig 1 is said to cross-cut groundwater flow systems (Line 42), but it is difficult to see this in Fig 1. Fig 2 – again unclear as to what the point is that the authors are making – the pink arrows do not always show higher ΔD for a given $\Delta 18O$ – are the pink arrows general direction of movement, or movement of individual samples? There are so many points on the figure, that it is difficult to see what is happening – perhaps put all of the points on one figure, and then just the points that you say are moving post-EQ on another?

It is unclear what actually happened with regards to the measured groundwater levels – the description in the introduction is confusing. The first figure in the supplementary material helped me to understand, and should be within the main paper. But I would also like to see some example hydrographs. One of the main issues that is not mentioned or argued is that groundwater level fluctuate in response to many drivers – if the authors are arguing that groundwater levels are still elevated after a year, how have they excluded other reasons? Rutter et al (2017) using simple analytical modelling to show that the rise in GWLs was not being driver by climate or abstraction – I would have expected some similar evidence. The manuscript really needs to justify why any GWL changes were due to EQs – especially after a year – there are many other drivers that need to be accounted for.

There are contradictions or apparent contradictions throughout the paper. Because of the lack of clarity in lines 43-52, it appears that there is a contradiction as to whether water levels increased or decreased. Line 56 there is talk of crustal extension, then Line 59 talks about contraction. In Line 105 the authors say the shift is smaller for unconfined aquifers, then in line 126 say that the change is especially changed for the unconfined aquifers.

Line 55-58 – I got lost as to the point being made

Line 66, the authors introduce hot springs, without stating their relevance.

Line 68 – what second hypothesis? The manuscript needs to be clear rather than the reader having to go back and work out what it is talking about.

Lines 86-88 – it is difficult to follow the reasoning.

At various point there is mention of unconfined vs confined aquifers, but the figures do not differentiate between the two.

Lines 139-140 does not make sense

Line 184 the authors introduce 'stagnant' flow areas with no other introduction.

There are many other part where there is further explanation required.

Originality and significance:

It is unclear whether the conclusions are original as I was not able to follow enough of the reasoning.

Data & methodology:

The approach was theoretically sound, but the arguments were difficult to follow.

Appropriate use of statistics and treatment of uncertainties:
There is no mention of uncertainties.

Conclusions:

There were no conclusions presented. It was difficult to follow the reasoning, there were too many contradictions, and too little justification of statements to be able to justify the discussion.

Suggested improvements:

Much better explanation throughout, including a more robust introduction, and better presentation of the results would help.

Clarity and context:

The first half of the abstract was good – then I lost the reasoning and explanation of water level changes in the final part. There was no introduction, and no conclusions, and the reasoning throughout was difficult to follow.

Reviewer #3 (Remarks to the Author):

Re.: review: "Earthquakes release water from mountains", by Hosono et al.

This manuscript by Hosono et al. discusses the groundwater hydrology response to a large earthquake in Japan. The findings of the manuscript are based on an unprecedented large stable water isotope database covering all major units of the hydrological system from before and after the 7 Mw Kumamoto earthquake. Based on the analysis of the dataset the authors can demonstrate a major change of groundwater system for the entire area. I find in particular interesting the demonstration of changes between the recharge and discharge area that have major implications for our understanding of subsurface flow pathways on the landscape scale, from mountains to lowlands. The findings are well suited to be published in Nature Communications. I like to congratulate the authors for this very unique and insightful work.

I have three minor to major points and several editorial comments.

1) Although the authors try to rule out in detail the potential seasonality effect on the isotopic signature, I am not yet entirely convinced by the arguments and documentation. The main problem is that all pre-earthquake samples represent the low water season and a good share of the post-earthquake samples do cover the high-water season (July and August) and the months after (September and October). Especially in the end of the rainy season, groundwater should reflect the purging of the aquifer and might potentially inherit seasonality signatures. This naturally asks the question of seasonality effects on the observations. I agree with the authors that groundwater should be well mixed and have an overall constant signature, however, I find this should be demonstrated more solid. In the reference 35 that was cited to underline the absence of seasonality in groundwater I could not find any isotopic data, maybe the authors include referenced data in the suppl. materials. I suggest to plot all groundwater data (incl. springs) in the dD vs $18O$ space as in figure 2 but color coded by the day of the year of sampling. Doing this should clearly demonstrate that the observed shifts towards mountain water signatures with less evaporation signals are real.

2) In order to underline the physics behind the interpretations, the authors should provide some back of the envelope estimations of how fast the waters have to travel over distance in order to

explain the changes of the isotopic signatures? I find this is important to first demonstrate the physical plausibility and second might even provide more evidence about the subsurface flow pathways. E.g. underpin the interpretation of predominant flow along fractures, cracks or any other large conduits.

3) My last point refers to the groundwater increase in the recharge areas. I find the argumentation by the authors quite convincing but was wondering where the extra water is coming from or let's say is missing? I understand that increased infiltration capacity in the recharge areas leads to a more effective groundwater recharge, leading to a rising of the groundwater table. However, if a substantial amount of the water is additionally infiltrated, or displaced, to the groundwater after the earthquake some other part of the hydrological system must receive less water. Was in the mountain areas any drop in river discharge and groundwater observed that might explain the overall water budget? I find confusing to not refer to the mountains as recharge area. What you demonstrate is that the recharge from the mountains to the lowlands has actually increased. The zone you term as recharge area is more a storage area, recharge to the groundwater is clearly higher from the mountains.

Comments:

- Line 61 to 63: I find this is a valid argument but this would imply that the deep water is fast upwelling. What about a more dynamic response where deeper water is pushing first the overlying shallower water out?
- Line 68-69: The hypotheses need to be better outlined. It is not really clear what the second hypothesis is.
- Line 74: To avoid confusion with be precise on the water level rise, "ground water level rise".
- Line 112: No isotope data is reported in the reference.
- Line 139: Fig. 4a should be suppl. material? There is no figure 4 in main text.
- Line 204-205: I am not entirely sure that I understand this argument. The recharge area and elevated areas (mountains) should be the same? This seems to be a problem of definition. Maybe the separation between High mountain areas and recharge area is confusing.
- Figure 1: The Figure 1 a-b is borrowed wholesale from a different publication in WRR (eventually a reprint permission is needed?). The topographic information of a and c is redundant and I am not sure the volumetric strain changes (in b) are needed here. I would suggest to underline one map with an spatial interpolation of the water level change, e.g. after 45 days. One topographic map could be replaced by an isoscape map that actually illustrates the spatial distribution of isotopic signatures.
- Figure 2: I find it visually very difficult to understand the differences between c, b, and d. I suggest to focus only on the compartments that have changed and keep the others in a bigger overview plot. I don't understand the black arrows with the hot spring samples. Can the authors add the global meteoric water line for reference?

Response to Reviewers' Comments

Reviewer #1:

Major comments

I have two major comments, one concerning interpretation of the data and one concerning the disposition of the manuscript:

Reply: We appreciate very much the reviewer's interest in the topic covered by our manuscript, and we also thank the reviewer for the important comments that should be followed. We provide below our reply to each reviewer's individual comment.

Concerning interpretation of the data: Data in Fig. 2 are used to argue that “the isotope ratios of groundwaters changed ... from their original compositions towards compositional fields of mountain spring waters ...” (lines 125-128). What is actually observed on the figure is a clustering of isotopic data towards the compositional field of mountain foot spring waters. Prior to the earthquake a good proportion of groundwater data overlapped mountain spring data. This difference does not undermine the conclusions of the study: the wider spread of isotopic values prior to the earthquake probably affects mixing of mountain spring water, soil pore waters and surface river waters. After the earthquake, mountain foot spring water clearly dominates the system, supporting the authors' conclusions.

Reply #1-1: We agree. We have revised sentences explaining observed isotopic changes and their interpretations on lines, i.e., 135-141, 175-180.

Lines 135-141: In Fig. 3 and Supplementary Fig. 5, the most remarkable post-seismic isotopic changes are observed for groundwaters: the compositional range of samples changed from wider (shown by the field surrounded by black line in Fig. 3b-d and Supplementary Fig. 5) to a more narrow range that is similar to mountain foot spring waters before the earthquake (most red-circle symbols plot within a field surrounded by the blue line), regardless the sampling season, aquifer type (confined vs. unconfined), and area of aquifers (except the stagnant area).

Lines 175-180: The most remarkable result of our isotopic comparisons is that the composition of groundwaters changed from resembling a mixture of multiple sources into a composition with a signature identical to mountain foot spring waters. Water compositions did not change towards those of soil porewaters, river waters, and deep fluids (Fig. 3b-d, Supplementary Fig. 5). This result clearly indicates an increased contribution of water from mountain aquifers.

Concerning the disposition of the manuscript: Lines 55-80 are not results of this study: they are results of the study published in Water Resources Research by Hosono et al. (2019). These “results” should therefore form part of the previous section which should be entitled “Introduction” or similar. The results for the manuscript are described in lines 82-123, so this section should be entitled “Results”. These new results are discussed in lines 125-194, so this section should be entitled “Discussion”. The conclusion and wider significance of the study are presented in lines 197-216, so this section should also be entitled accordingly.

Reply #1-2: We agree. We have reconstructed the manuscript throughout following reviewer's comment. Contents in Results and Discussion are briefly stated in the last paragraph of the Introduction. Conclusions are included in the later part of Discussion on lines 272-293 (this journal doesn't have Conclusion section).

Minor comments

Lines 85-86: I suggest rewording “plot along near the local meteoric line during high water season (April to September including rainy season) but to the right to this line” to “plot slightly to the right of the local meteoric water line for the high-water season (April to September including rainy season).”

Reply #1-3: We agree. We have corrected this statement as suggestion on lines 102-103.

Lines 88 and 91: Here you refer to isotopic shifts due to partial evaporation and recharge elevation. This would be easier for the reader to follow by adding corresponding arrows to show the directions of these shifts in Fig. 2.

Reply #1-4: We agree. We have added corresponding arrows to show the compositional shift due to evaporation and recharge elevation in Fig. 2. Explanations were added on lines 103-109.

Lines 103-109: We thus suggest that these waters are recharged by precipitation during high-water season and were partly evaporated before infiltration (see evaporation trend shown by dotted arrow in Fig. 2)²³. In contrast, the samples of mountain springs and the Shira River plot to the left of the local meteoric water line. In addition, the high-elevation mountain springs and the Shira River waters generally show more depleted isotopic signatures than mountain foot springs (Fig. 2), reflecting their higher recharge elevations (see trend of altitude effect shown by solid arrow in Fig. 2).

Line 89: I suggest replacing “to the local meteoric water” with “of this”.

Reply #1-5: Thank you for the comment. We think “to the left of the local meteoric water line” is good at here in revised manuscript (line 106).

Line 94: I suggest replacing “during” with “for the”.

Reply #1-6: Corrected (line 111).

Line 97: I suggest replacing “are” with “could be”. It is clear that that “groundwaters in the study area are recharged by precipitation during April to September” but unclear that mixing of all three of soil porewaters, mountain aquifer waters and the Shira River waters must have occurred.

Reply #1-7: Corrected (line 114).

Lines 98-99: You state that the isotopic signatures of water are the same in unconfined and confined aquifers. I am not sure how the reader is expected to see this from the data as it is presented.

Reply #1-8: Thank you for the comment. We have provided new diagram comparing isotopic compositions between unconfined and confined aquifers in Supplementary Fig. 4a that supports statement here (lines 115-117).

Line 104: Please replace “mountain spring waters” with “mountain foot spring waters”

Reply #1-9: Corrected (line 177).

Line 139: Please replace “Fig. 4a” with “Supplementary Fig. 4a”.

Reply #1-10: Thank you for your correction. This mistake was corrected (line 202, Supplementary Fig. 5 in revised manuscript).

Reviewer #2:

Validity:

In general I consider that, in an attempt to keep the paper brief, there is a lot of background information left out, which makes interpretation of the paper difficult. The structure also makes it difficult to follow. As it is, I would say that the information presented is not sufficient to justify the proposed conclusions, but it would be interesting if the data could support the conclusions. There are also a number of specific issues:

Reply: Thank you for the important suggestions. In revised manuscript we have provided a robust Introduction (words count increased 268%) adding more background information in the main body (e.g., lines 40-76). More hydrogeological background was added also in the Methods (hydrogeological setting) with an additional cross-sectional figure in Supplementary Fig. 1. Moreover, we have restructured the manuscript and added more explanation throughout the main body (words count increased 144% in the main text). We have added the title Introduction (line 28) followed by Results (from line 98 to 220) and Discussion (line 222 to 293). Contents in Results and Discussion are briefly stated in the last paragraph of the Introduction. Conclusions are included in the later part of Discussion in lines 172-193 (this journal doesn't have Conclusions section). We believe these revisions increased validity and readability of the manuscript. We provide below our reply to each reviewer's individual comment.

The hypothesis is that an increase in permeability has resulted in a groundwater level rise. Increases in permeability cause a decrease in groundwater level (Darcy's Law). If the hypothesis is that there is enough additional input to overcome increased output, or is it is that permeability has increased in some areas and not in others, this needs to be clearly explained.

Reply #2-1: Permeability has increased in mountain aquifers due to seismic rupturing (please see concentrated rupture systems in the mountains shown in Fig. 1b) that led water release from mountains to downslope aquifers in regional groundwater flow systems. This is explained in lines 223-265. A large volume of waters could be stored in recharge areas since these areas consist of major groundwater reservoirs in Kumamoto called groundwater pools (Fig. 4a) (lines 248-249). Increases in permeability in mountain aquifers were confirmed by the observation that some springs in the mountains became dry after the earthquake (lines 228-229).

The figures are unclear about what they are showing. References to the figures say that they show something, but it is difficult to understand the point that is being made. For example, Fig 1 is said to cross-cut groundwater flow systems (Line 42), but it is difficult to see this in Fig 1. Fig 2 – again unclear as to what the point is that the authors are making – the pink arrows do not always show higher delta D for a given delta 18O – are the pink arrows general direction of movement, or movement of individual samples? There are so many points on the figure, that it is difficult to see what is happening – perhaps put all of the points on one figure, and then just the points that you say are moving post-EQ on another?

Reply #2-2: We agree. We have revised Fig. 1b considering the reviewer's comments, directly showing how rupture systems crosscut original groundwater flows. We have also revised sentences explaining observed isotopic changes shown in Fig. 3 (which has been revised from original version Fig. 2) and their interpretations, i.e., lines 135-141, 175-180.

Lines 135-141: In Fig. 3 and Supplementary Fig. 5, the most remarkable post-seismic isotopic changes are observed for groundwaters: the compositional range of samples changed from wider (shown by the field surrounded by black line in Fig. 3b-d and Supplementary Fig. 5) to a more narrow range that is similar to mountain foot spring waters before the earthquake (most red-circle symbols plot within a field surrounded by the blue line), regardless the sampling season, aquifer type (confined vs. unconfined), and area of aquifers (except the stagnant area).

Lines 175-180: The most remarkable result of our isotopic comparisons is that the composition of groundwaters changed from resembling a mixture of multiple sources into a composition with a signature identical to mountain foot spring waters. Water compositions did not change towards those of soil porewaters, river waters, and deep fluids (Fig. 3b-d, Supplementary Fig. 5). This result clearly indicates an increased contribution of water from mountain aquifers.

It is unclear what actually happened with regards to the measured groundwater levels – the description in the introduction is confusing. The first figure in the supplementary material helped me to understand, and should be within the main paper. But I would also like to see some example hydrographs. One of the main issues that is not mentioned or argued is that groundwater level fluctuate in response to many drivers – if the authors are arguing that groundwater levels are still elevated after a year, how have they excluded other reasons? Rutter et al (2017) using simple analytical modelling to show that the rise in GWLs was not being driver by climate or abstraction – I would have expected some similar evidence. The manuscript really needs to justify why any GWL changes were due to EQs – especially after a year – there are many other drivers that need to be accounted for.

Reply #2-3: Thank you very much for important suggestions. Hydrographs are added in Supplementary Fig. 3. Recent hydrological analysis, with a similar concept to Rutter et al (2017), has revealed that the water level increase was triggered coseismically, peaked in 4-5 months, and persists at least 3 years after the main shock based on water balance modeling that considered all other possible factors to cause water level changes (i.e., climate and artificial water uptake and recharge) (lines 72-76). More background information on groundwater fluctuations and coseismic water level changes are provided in lines 59-76, 247-249, and 296-319 (Methods).

There are contradictions or apparent contradictions throughout the paper. Because of the lack of clarity in lines 43-52, it appears that there is a contradiction as to whether water levels increased or decreased. Line 56 there is talk of crustal extension, then Line 59 talks about contraction. In Line 105 the authors say the shift is smaller for unconfined aquifers, then in line 126 say that the change is especially changed for the unconfined aquifers.

Reply #2-4: We have revised sentences considering the comments. Statement about crustal contraction was deleted from the text. Isotopic change features are similar for both aquifers (line 140) and we have deleted other statements from the text.

Line 55-58 – I got lost as to the point being made

Reply #2-5: We believe now these sentences (lines 77-79) are more readable with newly added background information on lines 40-76.

Line 66, the authors introduce hot springs, without stating their relevance.

Reply #2-6: Statements about hot springs were deleted from the Introduction. Sampling strategy (line 94) and detailed results are provided in the Methods (line 346-349) and Results (lines 147-158), respectively.

Line 68 – what second hypothesis? The manuscript needs to be clear rather than the reader having to go back and work out what it is talking about.

Reply: We have deleted this statement from the text.

Lines 86-88 – it is difficult to follow the reasoning.

At various point there is mention of unconfined vs confined aquifers, but the figures do not differentiate between the two.

Reply: We agree. We have added figures showing isotopic comparisons between unconfined and confined aquifers in Supplementary Fig. 4a,c,d, and 5a, in addition to Fig. 3 in the main text.

Lines 139-140 does not make sense

Reply: We have deleted these sentences from the text.

Line 184 the authors introduce ‘stagnant’ flow areas with no other introduction.

There are many other part where there is further explanation required.

Reply #2-7: Information on the hydrological setting was provided on lines 40-58 including groundwater flows in the stagnant area (line 51-52).

Lines 51-52: Some nearly stagnant groundwaters remain in the plains and coastal regions to the west of Lake Ezu (stagnant area).

Originality and significance:

It is unclear whether the conclusions are original as I was not able to follow enough of the reasoning.

Data & methodology:

The approach was theoretically sound, but the arguments were difficult to follow.

Conclusions:

There were no conclusions presented. It was difficult to follow the reasoning, there were too many contradictions, and too little justification of statements to be able to justify the discussion.

Clarity and context:

The first half of the abstract was good – then I lost the reasoning and explanation of water level changes in the final part. There was no introduction, and no conclusions, and the reasoning throughout was difficult to follow.

Suggested improvements:

Much better explanation throughout, including a more robust introduction, and better presentation of the results would help.

Reply: Thank you for important suggestions. Considering reviewer's claims and suggestions, we have provided a robust Introduction (words counts increased 268%) and added more explanation throughout the main body (words counts increased 144% in the main text). We have added more explanation in Methods about the hydrogeological setting. All figures are carefully revised. We believe these revisions significantly increased validity and readability of the manuscript. Conclusions are included in the later part of Discussion in lines 172-193 (this journal doesn't have Conclusions section).

Appropriate use of statistics and treatment of uncertainties:

There is no mention of uncertainties.

Reply #2-8: Analytical errors were explained in the Methods section (lines 357-359 and 366-367) and included in all figures.

Reviewer #3:

This manuscript by Hosono et al. discusses the groundwater hydrology response to a large earthquake in Japan. The findings of the manuscript are based on an unprecedented large stable water isotope database covering all major units of the hydrological system from before and after the 7 Mw Kumamoto earthquake. Based on the analysis of the dataset the authors can demonstrate a major change of groundwater system for the entire area. I find in particular interesting the demonstration of changes between the recharge and discharge area that have major implications for our understanding of subsurface flow pathways on the landscape scale, from mountains to lowlands. The findings are well suited to be published in Nature Communications. I like to congratulate the authors for this very unique and insightful work.

I have three minor to major points and several editorial comments.

Reply: We appreciate very much the reviewer's interest in the topic covered by our manuscript, and we also thank the reviewer for the important comments that should be followed. We provide below our reply to each reviewer's individual comment.

1) Although the authors try to rule out in detail the potential seasonality effect on the isotopic signature, I am not yet entirely convinced by the arguments and documentation. The main problem is that all pre-earthquake samples represent the low water season and a good share of the post-earthquake samples do cover the high-water season (July and August) and the months after (September and October). Especially in the end of the rainy season, groundwater should reflect the purging of the aquifer and might potentially inherit seasonality signatures. This naturally asks the question of seasonality effects on the observations. I agree with the authors that groundwater should be well mixed and have an overall constant signature, however, I find this should be demonstrated more solid. In the reference 35 that was cited to underline the absence of seasonality in groundwater I could not find any isotopic data, maybe the authors include referenced data in the suppl. materials. I suggest to plot all groundwater data (incl. springs) in the δD vs $\delta^{18}O$ space as in figure 2 but color coded by the day of the year of sampling. Doing this should clearly demonstrate that the observed shifts towards mountain water signatures with less evaporation signals are real.

Reply #3-1: Thank you very much for very important comments. Considering the reviewer's comments, we have added more analysis on the seasonality of isotopic signatures for collected water samples. We have added new figures (Supplementary Fig. 4b-d) showing compositional differences between higher and lower water seasons for all types of water samples (spring, unconfined and confined ground waters). We have added more explanations on lines 124-131. Moreover, we have provided isotopic data of groundwaters presented in previous work (reference 34, replaced with reference 35 in original version manuscript) in Supplementary Table 3, showing their stabilities throughout the year.

Lines 124-131: Accordingly, isotopic compositions of spring water samples did not show systematic changes due to different sampling seasons (Supplementary Fig. 4b). In addition, groundwater samples collected during two different seasons show overlapping isotopic compositions for both aquifers (Supplementary Fig. 4c,d). However, there are some samples from the higher water season (October to December) with greater $\delta^{18}\text{O}$ than those for the lower water season (January to March) (Supplementary Fig. 4c,d). This may be due to a lack of samples from the lower water season or may imply relatively higher contribution of soil porewaters transported through preferential flow pathways in the rainy season, as may be typical in recharge areas³⁵.

2) In order to underline the physics behind the interpretations, the authors should provide some back of the envelope estimations of how fast the waters have to travel over distance in order to explain the changes of the isotopic signatures? I find this is important to first demonstrate the physical plausibility and second might even provide more evidence about the subsurface flow pathways. E.g. underpin the interpretation of predominant flow along fractures, cracks or any other large conduits.

Reply #3-2: These comments are also important issues to be addressed. We have shown more about time scale of groundwater flows on lines 72-76, 244-250 and 251-260 in the Discussion.

Lines 72-76: Recent hydrological analysis²⁸, based on water balance modeling that considered all other possible factors to cause water level changes (i.e., climate and artificial water uptake and recharge), revealed that the water level increase was triggered coseismically, peaked in 4-5 months, and persists at least 3 years after the main shock.

Lines 244-250: The response of mountain waters appeared within 1 day as increased river and spring discharges^{6,47} and abnormal groundwater level rise in recharge area (Supplementary Fig. 3). The water levels in this area continuously rose (Supplementary Figs. 2 and 3) and peaked 4-5 months after the main shock with a maximum abnormal water level rise of $\sim 11\text{ m}^{28}$. In general, the recharge areas consist of major groundwater reservoirs in Kumamoto called groundwater pools (Fig. 4a) and have the highest seasonal water level fluctuations of up to $\sim 10\text{ m}^{25}$. Thus a large volume of coseismically-released mountain waters can be added to the recharge area.

Lines 251-260: The isotopic results further suggest that increased water in the groundwater pools was then transported down slope and led to water level recovery in lateral flow to recharge areas within the annual hydrologic cycle (Fig. 3d and Supplementary Fig. 2). This isotopic constraint requires a faster flow than that suggested by water residence times of a few to a few 10 years estimated by chemical age tracers²⁵. Aquifers in recharge and discharge areas and partly in lateral flow areas were also subject to permeability enhancement associated with the formation of rupture systems (Fig. 1b)²⁶. In addition, previous studies identified faster flow appearing only during heavy rain³⁵. We can therefore assume that the dropped water levels

recovered by the contribution of large amounts of mountain waters (ca. 10^8 m^3)⁴⁷ that were transported through these preferential pathways.

3) My last point refers to the groundwater increase in the recharge areas. I find the argumentation by the authors quite convincing but was wondering where the extra water is coming from or let's say is missing? I understand that increased infiltration capacity in the recharge areas leads to a more effective groundwater recharge, leading to a rising of the groundwater table. However, if a substantial amount of the water is additionally infiltrated, or displaced, to the groundwater after the earthquake some other part of the hydrological system must receive less water. Was in the mountain areas any drop in river discharge and groundwater observed that might explain the overall water budget? I find confusing to not refer to the mountains as recharge area. What you demonstrate is that the recharge from the mountains to the lowlands has actually increased. The zone you term as recharge area is more a storage area, recharge to the groundwater is clearly higher from the mountains.

Reply #3-3: Thank you very much for important suggestions. We agree with the reviewer's points. Although there is not much direct evidence showing water level decrease in mountain aquifers due to lack of monitoring wells inside mountains, we have added some descriptions of a water level drop observed in mountain areas in lines 228-229. Other evidence of mountain water release is also provided on lines 204-206 and 282-283.

Lines 228-229: This hypothesis is consistent with the observation that some springs in the mountains became dry after the earthquake³⁶.

Lines 204-206: This result supports the hypothesis derived from hydrological analyses both for surface and subsurface waters^{6,28} and hydrochemical signatures that suggest mixing of diluted mountain aquifer waters³⁸.

Lines 282-283: Numerical simulation involving permeability changes has reproduced these regional flow changes using a physically based integrated watershed modeling tool⁴⁷.

For issues of areal definition used in this study, we have provided in the revised manuscript a robust Introduction (words count increased 268%) adding more background information in the main body (e.g., lines 40-76). More hydrogeological background was added also in the Methods (hydrogeological setting) and Supplementary Fig. 1. Moreover, we have restructured the manuscript and added more explanations throughout the main body (words count increased 144% in the main text). We believe that these revisions increased validity and readability of the manuscript.

Comments:

Line 61 to 63: I find this is a valid argument but this would imply that the deep water is fast upwelling. What about a more dynamic response where deeper water is pushing first the overlying shallower water out?

Reply #3-4: We have deleted these sentences from the text. For the hypothesis that deeper water is pushing first the overlying shallower water out, we did not detect signals of deep fluid upwelling in deep aquifers except localized areas, e.g., near epicenters. We have added more explanation about deep fluid upwelling with recent citations on lines 181-197.

Lines 181-197: Recent analyses using chemical³⁸ and microbiological tracers³⁹ have detected in some localized areas an increased contribution of soil porewaters, river waters and deep fluids after the occurrence of the 2016 Kumamoto earthquake. The study also revealed

post-seismic increase in contribution of deep fluids upwelling to the surface aquifers based on geochemical markers, i.e., Cl⁻, SO₄²⁻, and B, as has been documents in many other instances⁴⁰⁻⁴⁵. This phenomenon occurs near the epicentre of earthquakes and regions with geothermal fields³⁸. These hydrochemical and microbial anomalies continued for at least 2 years after the earthquake. However, these effects are not reflected in a change of groundwater isotopic compositions. Thus, their contributions are negligible in terms of water volume and unlikely to cause the observed groundwater level rise and recovery in regional groundwater flow systems, except for river water contribution that will be discussed later.

Line 68-69: The hypotheses need to be better outlined. It is not really clear what the second hypothesis is.

Reply: We agree. We have deleted this mention.

Line 74: To avoid confusion with be precise on the water level rise, “ground water level rise”.

Reply #3-5: We agree. We have rephrased “water level rise” with “groundwater level rise” on Line 85.

Line 112: No isotope data is reported in the reference.

Reply #3-6: Thank you for important suggestion. We have cited the correct reference (reference 34) in line 123 and also provided their isotopic data in Supplementary Table 3.

Line 139: Fig. 4a should be suppl. material? There is no figure 4 in main text.

Reply #3-7: Thank you for your correction. We corrected our mistake (line 202, Supplementary Fig. 5 in revised manuscript).

Line 204-205: I am not entirely sure that I understand this argument. The recharge area and elevated areas (mountains) should be the same? This seems to be a problem of definition. Maybe the separation between High mountain areas and recharge area is confusing.

Reply #3-8: Thank you for important comment. As mentioned in reply to previous comments, we have added more background information in the main body of the revised manuscript (e.g., lines 40-76). More hydrogeological background was added also in the Methods (hydrogeological setting) and Supplementary Fig. 1. We believe these revisions increased readability of the manuscript.

Figure 1: The Figure 1 a-b is borrowed wholesale from a different publication in WRR (eventually a reprint permission is needed?). The topographic information of a and c is redundant and I am not sure the volumetric strain changes (in b) are needed here. I would suggest to underline one map with an spatial interpolation of the water level change, e.g. after 45 days. One topographic map could be replaced by an isoscape map that actually illustrates the spatial distribution of isotopic signatures.

Reply #3-9: Thank you for important suggestion. We have revised Fig. 1 taking this comment and also comments provided from other reviewers. We still think that showing isoscape map including data from different water bodies from different sampling events is difficult and not efficient but may important for providing whole picture of the sampling scale and strategy

which is one of major new contributions of this work. Thus we still show map displaying sampling locations in Fig. 1c.

Figure 2: I find it visually very difficult to understand the differences between c, b, and d. I suggest to focus only on the compartments that have changed and keep the others in a bigger overview plot. I don't understand the black arrows with the hot spring samples. Can the authors add the global meteoric water line for reference?

Reply #3-10: Thank you for important suggestion. We have revised Figs. 2 and 3 taking this comment and also comments provided from other reviewers. Black arrows with the hot spring samples mean that sample plots are out of range. To make it clear we have shown their isotopic ratios besides the plots and changed the position of the arrows. We have added the global meteoric water line in Fig. 2.

Reviewers' comments:

Reviewer #1 (Remarks to the Author):

Second review by Alasdair Skelton of Earthquakes release water from mountains submitted to Nature Communications by Takahiro Hosono and co-authors.

I am fully satisfied that the authors have responded adequately to all of my comments and I recommend publication of this manuscript in its present form.

Reviewer #2 (Remarks to the Author):

I think this is a useful paper which attempts to answer an important question - whether there is any evidence for a change in recharge to an aquifer system after a major EQ. Other papers have suggested this as a possibility, but the large number of isotope samples provides possibly more compelling evidence for this.

However, overall, I still find the paper confusing and numerous things need to be clarified. Some of this is confusion about the geographic area and some of the terminology used - but then there should be no assumption that the reader will be familiar with the area, so the paper should be clear.

I also feel that the paper puts forward some statements that, to me, are still hypotheses. None of this can be proved - what the paper is doing is putting forward evidence that suggest there is something happening, so to me it should be framed more as interpreting data to test a hypothesis, and less as a certain outcome.

I find the order of parts of the paper leads to confusion - there are some basic background observations half way through the paper that would help the reader understand the situation if explained earlier on.

The main point is that groundwater appears to have moved closer to mountain foot spring water in terms of composition. However, before the EQ, there was already considerable overlap between groundwater and mountain spring water. Post-EQ, it appears that the composition of some groundwater samples has moved towards that of mountain spring water, but the majority were already pretty similar. I don't think this point is brought out. Effectively there is a subtle but observable shift away from the soil porewater composition, towards the mountain foot spring composition. You can't say that post-EQ the distributions are identical, though they become more similar.

General comments:

I got confused throughout about "high water", "higher water" and "lower water" seasons. At 127 you say the "higher water season" is Oct - Dec, on the page before you say the "high water season" is Apr-Sept. It would help many people to understand what you mean by these descriptions (is there a very strong seasonal signal that is greater than any longer-term variability?), and when the different seasons are, as there seems to be some inconsistency. Personally I have not come across groundwater-focussed papers using this terminology before. I find the order of the sections a bit odd - but this may be the preferred order? That is Method and Samples after discussion.

Specific comments:

Line 64 - I would say that this is still a hypothesis

Lines 73-76 - needs re-wording - the sentence just doesn't read well and I am not clear about the

point being made.

Lines 84-85 – be clear that you are talking about a possible increase in permeability in the recharge areas and GWL increase in the down-gradient regional GW flow system

Line 113 – you need to be clear here, and throughout the paper where you are referring to. When you say the study area, which bit of the study area do you mean?

line 141 – where are the stagnant groundwaters on the figure?

Line 148 – I can only see 2 groups on Fig 3a, not 4 – this may be down to the size of the figure

Line 161 – they don't compare samples from different seasons – the point of these figures is to compare before and after the EQ

Line 171 – this is hypothesised to have caused post-seismic water levels to rise

Line 177 – they have become more similar – they are not identical

Line 181 onwards – I would put this paragraph possibly after line 220, and start it with "in contrast to the previous evidence...". You have been putting forward one hypothesis, then this is in disagreement with that hypothesis, so you need to be clear that this doesn't fit with your hypothesis. Why do you say that this effect is localised? Is it that the evidence doesn't support it in a widespread area – or is there just not enough data?

Line 218 – similar to, not identical

Line 239 – it doesn't demonstrate, it suggests

Line 248/9 – you introduce "groundwater pools" for the first time. These need some explanation.

249-250 – this doesn't follow on from the previous sentence – why could you add a large volume?

Line 252-253 – but the GW pools are in the recharge area

Line 257 – this point doesn't appear to be "in addition"

Lines 257-260 – this doesn't make sense to me. This is the first mention of preferential pathways, and if you are going to talk about them, then there should be some introduction much earlier

Lines 262-263 – I am not sure what preferential pathways surface waters would travel along. A surface water itself obviously travels along a preferential pathway, but are you talking about recharge to groundwater?

Line 269 – you don't need to have liquefaction to get a co-seismic increase in pore pressure

Line 270 – the results support the previously proposed scenario – of what?

Line 272 onwards – some of this needs to be introductory, somewhere around Line 62 – that would make some of the introductory information much more understandable

Line 282 – 283 – if this paper supports this isotope work, then it is really important to make that clear.

Line 283 – I am getting quite confused about which water levels

Lines 285 – 286 – does the modelling talked about just previously, provide any evidence that you would get a long-term rise in GWLs as a result of permeability increase in the mountain aquifers? I would have thought that the system would return to normal – there is no change in recharge presumably, so if it resulted in a long term increase in groundwater levels down-gradient, then some water must have gone elsewhere

Line 287-288 – you say earlier in the paper that some of the lines of evidence basically disagree with the isotope results, and put the contradictions down to localised effects.

Line 289 – it shows how earthquakes may alter regional systems

Line 290 – first time there is mention of high storage. In many volcanics, storage is low due to lack of connected porosity. Permeability relies on interconnected fractures/joints.

Line 315 – say May and June 2016 to make it clear

Fig 1 and Supplementary Fig 8 seem to have different areas identified as the recharge/significant GWL rise areas

Supplementary Fig 2 – I cannot read the key at all

Supplementary Fig 3 – the groundwater level change is in relation to what? The instantaneous water level at some point before the EQ?

Reviewer #3 (Remarks to the Author):

Re. Hosono et al.

The authors I have done a great job revising the manuscript. I see that all my comments and concerns have been answered and especially the figures are much more clear now. I have no further comments and recommend the manuscript for publication in Nature communications.

Sincerely Christoff Andermann

Response to Reviewers' comments

Reviewers' comments:

Reviewer #1 (Remarks to the Author):

Second review by Alasdair Skelton of Earthquakes release water from mountains submitted to Nature Communications by Takahiro Hosono and co-authors.

I am fully satisfied that the authors have responded adequately to all of my comments and I recommend publication of this manuscript in its present form.

Reviewer #2 (Remarks to the Author):

I think this is a useful paper which attempts to answer an important question - whether there is any evidence for a change in recharge to an aquifer system after a major EQ. Other papers have suggested this as a possibility, but the large number of isotope samples provides possibly more compelling evidence for this.

However, overall, I still find the paper confusing and numerous things need to be clarified. Some of this is confusion about the geographic area and some of the terminology used – but then there should be no assumption that the reader will be familiar with the area, so the paper should be clear.

I also feel that the paper puts forward some statements that, to me, are still hypotheses. None of this can be proved - what the paper is doing is putting forward evidence that suggest there is something happening, so to me it should be framed more as interpreting data to test a hypothesis, and less as a certain outcome.

We added “possibly” on line 66; we used “possible” on line 84; we removed “It is clear” on line 113; added “These may have” to line 171; changed “clearly indicates” on line 178 to “implies”; added “may” to line 287.

I find the order of parts of the paper leads to confusion – there are some basic background observations half way through the paper that would help the reader understand the situation if explained earlier on.

The main point is that groundwater appears to have moved closer to mountain foot spring water in terms of composition. However, before the EQ, there was already considerable overlap between groundwater and mountain spring water. Post-EQ, it appears that the composition of some groundwater samples has moved towards that of mountain spring water, but the majority were already pretty similar. I don't think this point is brought out. Effectively there is a subtle but observable shift away from the soil porewater composition, towards the mountain foot spring

composition. You can't say that post-EQ the distributions are identical, though they become more similar.

On line 176 we changed "identical" to "more similar".

General comments:

I got confused throughout about "high water", "higher water" and "lower water" seasons. At 127 you say the "higher water season" is Oct – Dec, on the page before you say the "high water season" is Apr-Sept. It would help many people to understand what you mean by these descriptions (is there a very strong seasonal signal that is greater than any longer-term variability?), and when the different seasons are, as there seems to be some inconsistency. Personally I have not come across groundwater-focussed papers using this terminology before.

Thank you for the comment. We have added definition of terminology "higher-water season" and "low-water season" in the Methods section (lines 299-301). Other expressions "higher water" and "lower water" seasons are deleted from the text throughout.

I find the order of the sections a bit odd – but this may be the preferred order? That is Method and Samples after discussion.

This is odd (non-traditional), but it the style of *Nature Communications*.

Specific comments:

Line 64 – I would say that this is still a hypothesis

We are citing already-published papers here that drew these conclusions. We thus added the word "possibly" (line 66).

Lines 73-76 – needs re-wording – the sentence just doesn't read well and I am not clear about the point being made.

Reworded to

"Hydrological models based on water budgets²⁸ that considered other possible factors that might cause water level changes (e.g., climate and artificial water uptake and recharge), revealed that the water level increase was triggered coseismically, peaked in 4-5 months, and persisted at least 3 years after the main shock." (lines 73-76)

Lines 84-85 – be clear that you are talking about a possible increase in permeability in the recharge areas and GWL increase in the down-gradient regional GW flow system

Changed sentence to

"It is therefore possible that increased permeability in the upstream area is responsible for the observed groundwater level rise in downgradient groundwater flow systems." (lines 84-86)

Line 113 – you need to be clear here, and throughout the paper where you are referring to. When you say the study area, which bit of the study area do you mean?

We removed the expression “in the study area” here as it was not needed (line 113).

line 141 - where are the stagnant groundwaters on the figure?

We now refer readers to Figure 1a (line 141) which has the stagnant groundwaters labeled.

Line 148 – I can only see 2 groups on Fig 3a, not 4 – this may be down to the size of the figure

We now refer readers to Figure 3a at the proper location (line 148) to support the explanation ‘waters obtained from deep water reservoirs (180 to 1,300 m below the ground surface) show a wide range’.

Line 161 – they don’t compare samples from different seasons – the point of these figures is to compare before and after the EQ

We changed the word “compare” to “include” since the point is that we need to assess whether we are seeing seasonal differences (line 161).

Line 171 – this is hypothesised to have caused post-seismic water levels to rise

Added the word “may” (line 171).

Line 177 – they have become more similar – they are not identical

Changed as suggested (line 176).

Line 181 onwards – I would put this paragraph possibly after line 220, and start it with “in contrast to the previous evidence....”. You have been putting forward one hypothesis, then this is in disagreement with that hypothesis, so you need to be clear that this doesn’t fit with your hypothesis. Why do you say that this effect is localised? Is it that the evidence doesn’t support it in a widespread area – or is there just not enough data?

This paragraph was moved as suggested (lines 203-219).

Line 218 – similar to, not identical

Changed word to “approach” (line 200).

Line 239 – it doesn’t demonstrate, it suggests

Changed to “suggesting” (line 238).

Line 248/9 – you introduce “groundwater pools” for the first time. These need some explanation.
249-250 – this doesn’t follow on from the previous sentence – why could you add a large volume?

We added that the groundwater pools can be the source of water (lines 246-250).

Line 252-253 – but the GW pools are in the recharge area

We removed the word “increased” to convey that the groundwater pools are providing water to down slope regions (line 251).

Line 257 – this point doesn’t appear to be “in addition”

Removed the word “in addition” (line 257).

Lines 257-260 – this doesn’t make sense to me. This is the first mention of preferential pathways, and if you are going to talk about them, then there should be some introduction much earlier

Lines 262-263 – I am not sure what preferential pathways surface waters would travel along. A surface water itself obviously travels along a preferential pathway, but are you talking about recharge to groundwater?

We added the clarification of “subsurface” (line 259).

Line 269 – you don’t need to have liquefaction to get a co-seismic increase in pore pressure

We agree, but here there is very little groundwater flow (the stagnant region) so we need a local process. Regardless, here we are citing ref. 6 for this observation and interpretation.

Line 270 – the results support the previously proposed scenario – of what?

We removed the sentence (line 269).

Line 272 onwards – some of this needs to be introductory, somewhere around Line 62 – that would make some of the introductory information much more understandable

The challenge is that from line 270 to the end are conclusions based on the data and interpretations in the paper.

Line 282 – 283 – if this paper supports this isotope work, then it is really important to make that clear.

This cited paper did not simulate isotopic changes so we cannot make a more quantitative statement.

Line 283 – I am getting quite confused about which water levels

Agreed. We reversed the sentences so that we can clarify that it is downslope water levels that recovered (lines 281-284).

Lines 285 – 286 – does the modelling talked about just previously, provide any evidence that you would get a long-term rise in GWLs as a result of permeability increase in the mountain aquifers? I would have thought that the system would return to normal – there is no change in recharge presumably, so if it resulted in a long term increase in groundwater levels down-gradient, then some water must have gone elsewhere

Hopefully the rewording to address the previous comment clarifies where changes occurred.

Line 287-288 – you say earlier in the paper that some of the lines of evidence basically disagree with the isotope results, and put the contradictions down to localised effects.

Major findings from previous papers are consistent with the findings of the present paper except some localized effects which were not detected by isotopic signals.

Line 289 – it shows how earthquakes may alter regional systems

Added the word “may” as requested (line 287).

Line 290 – first time there is mention of high storage. In many volcanics, storage is low due to lack of connected porosity. Permeability relies on interconnected fractures/joints.

Added the word “often” (line 288) – in Kumamoto, the volcanic aquifers have high storage and permeability and this is true in many other settings with young volcanic rocks (pyroclastic or lava flows).

Line 315 – say May and June 2016 to make it clear

Changed as suggested (line 317).

Fig 1 and Supplementary Fig 8 seem to have different areas identified as the recharge/significant GWL rise areas

Supplementary Fig 8 now shows the area where isotopic data are available.

Supplementary Fig 2 – I cannot read the key at all

We increased the size of letters in Supplementary Fig 2.

Supplementary Fig 3 – the groundwater level change is in relation to what? The instantaneous water level at some point before the EQ?

Thank you for the comment. Here we mean ‘in relation to water level about 26 minutes before the foreshock’ and this is stated in figure caption.

Reviewer #3 (Remarks to the Author):

Re. Hosono et al.

The authors I have done a great job revising the manuscript. I see that all my comments and concerns have been answered and especially the figures are much more clear now. I have no further comments and recommend the manuscript for publication in Nature communications.

Sincerely Christoff Andermann

REVIEWERS' COMMENTS:

Reviewer #2 (Remarks to the Author):

Thanks for addressing my previous comments. Just a couple of minor edits that I can see to help clarify the message.

Line 260 - it would be clearer if you say "lowered water levels due to the contribution"....

Line 285, I would prefer that you are clear this is still a hypothesis. Perhaps add in a couple of words such as "after the main shock, hypothesised to be because of persistent permeability increase".....?

Reply to comments from Reviewer #2

Line 260 - it would be clearer if you say "lowered water levels due to the contribution"....

We have revised the sentence as suggested: '... lowered water levels due to coseismic groundwater drawdown recovered...' (Line 259)

Line 285, I would prefer that you are clear this is still a hypothesis. Perhaps add in a couple of words such as "after the main shock, hypothesised to be because of persistent permeability increase".....?

We have revised this sentence as suggested: '...after the main shock, hypothesised to be because of persistent permeability increase...' (Line 284).

Additional information

Citation information was updated.